# Supervised pulmonary tele-rehabilitation and individualized home-based pulmonary rehabilitation for patients with COPD, unable to participate in center-based programs. The protocol for a multicenter randomized controlled trial - the REPORT study

**Christina Nielsen**[1]*, **Nina Godtfredsen**[1,2], **Stig Molsted**[2,3], **Charlotte Ulrik**[1,2], **Thomas Kallemose**[4], **Henrik Hansen**[1,5]

**1** Respiratory Research Unit and Department of Respiratory Medicine, Copenhagen University Hospital-Hvidovre, Hvidovre, Denmark, **2** Institute of Clinical Medicine, University of Copenhagen, Copenhagen, Denmark, **3** Department of Clinical Research, North Zealand Hospital, Hillerod, Denmark, **4** Clinical Research Center, Copenhagen University Hospital- Hvidovre, Hvidovre, Denmark, **5** Department of Rehabilitation Sciences and Physiotherapy, University of Antwerp, Antwerp, Belgium

☯ These authors contributed equally to this work.
* christina.nielsen.07@regionh.dk

## Abstract

### Introduction

Chronic obstructive pulmonary disease (COPD) costs EURO 1.4 billion annually in health-care costs. Pulmonary rehabilitation (PR) is a vital aspect of care for patients with COPD, but despite the compelling evidence, it is delivered to less than 30%. Frequent transport to the center-based program is regularly reported as reasons for non-attendance. The effectiveness and feasibility of pulmonary tele-rehabilitation (PTR) and home-based pulmonary rehabilitation (HPR) have never been investigated in patients with COPD who are unable to attend conventional outpatient PR.

### Materials and methods

This study is a multicenter randomized controlled trial consisting of three parallel groups; PTR, HPR and a control group. 180 patients with moderate to very severe COPD, who are unable to attend in center-based PR programs will be included. The PTR group receives group-based resistance- and endurance training and patient education 60 min. twice a week for 10-weeks. HPR comprises an individual self-initiated home-based PR program with online motivational and professional counseling. The goal is to achieve at least 20 min. of muscle-endurance based exercises three days weekly for 10-weeks. The PTR and HPR group use a tablet with a conference system. The control group receives usual care (no PR). After completion of the intervention, the PTR and HPR groups are offered 65-weeks groupbased maintance program supervised once a week online via tablet. The primary

**Data Availability Statement:** Data access in Denmark are under strict juristic data protection law as imposed by the Danish Ministry of Justice. Any possible access or sharing demands a part application to; (1) Danish Data Protection Agency (email: dt@datatilsynet.dk), (2) Ethics Committee of the Capital Region (email: vek@regionh.dk), (3) National Health Data Authorities (email: kontakt@sundhedsdata.dk). Only if the applications are approved will the data be considered available for sharing.

**Funding:** This research project has received specific grants from the Danish Lung Foundation (Charitable funding), Telemedical Center Regional Capital Copenhagen (Governmental funding), TrygFonden (Charitable funding), Danish Association for Physiotherapists (Charitable funding), Jascha Fonden (Charitable funding), Skibsreder Per Henriksen, R og hustrus fond (Charitable funding), Amager-Hvidovre Hospital Forskningspulje (Governmental funding) and Lundbeck Fonden (Charitable funding). The Grants covers expenses conducting the trial, salary for the project employed, and the University fees for the PhD education for Christina Nielsen. I have uploaded funding letters from TrygFonden (for Henrik Hansen - last author) and for myself from AHH Forskningspulje 2023 and 2023 along with Skibsreder Per Henriksen og Hustrus Fond. The project has received financial funding. The funders had no role in study design, data collection and analysis, decision to publish, or preparation of the manuscript.

**Competing interests:** The authors have declared that no competing interests exist.

**Abbreviations:** AE, Adverse events; ACSM, American College of Sports Medicine; BMI, Body Mass Index; BODE index, Body-mass index, airflow Obstruction, Dyspnea and Exercise; BORG CR-10, Borg Category-Ratio range 0–10 scale; CRF, Case Report Form; CIMT, Center for IT and Medico-Technology; COPD, Chronic obstructive pulmonary disease; CAT, COPD Assessment Test; CON, Control group; CUA, Conduct cost-utility analysis; CONSORT, Consolidated Standards of Reporting Trials; DARN, Desire, Ability, Reason and Need principles; FEV1, Forced Expiratory Volume in one second; FITT-VP, Frequency-Intensity-Time-Type-Volume-Progression principles; FVC, Forced Vital Capacity; GOLD, Global Initiative for Chronic Obstructive Lung Disease; HGS, Hand-grip strength; HPR, Home-based pulmonary rehabilitation; kg., Kilograms; MP, Maintenance program; MET, Metabolic equivalent of task; MCID, Minimal clinical important difference; min., Minutes; MFI-20, Multidimensional Fatigue

outcome is change in respiratory symptoms measured with the COPD Assessment Test after 10-weeks (primary endpoint).

## Discussion

The study aims to test a possible equivalence between PTR and HPR and their superiority to controls on respiratory symptoms. The study will provide valuable insights into the effectiveness of new rehabilitation models and maintenance programs for patients with COPD. If the two new delivery models can reduce respiratory symptoms, patients with moderate to very severe COPD can participate in both home- or centerbased PR.

## Trial registration

The trial is registrered and approved by the Ethics Committee of The Capital Region of Denmark (H-22015777; 29.08.2022) and the Danish Data Protection Agency (P-2022-245-13101, 25.05.2022). The trial is registrered at ClinicalTrials.gov, identifier: NCT05664945 (23.12.2022).

## Introduction

Chronic obstructive pulmonary disease (COPD) is worldwide a common and debilitating disease. In Denmark, COPD costs approximately EURO 1.4 billion annually in citizens´ contacts and treatments within the Danish health care system [1]. It is estimated that 14.3% of all Danes aged 45 years or more of have COPD [2]. Improved treatment options and increasing life expectancy imply that the number of individuals with rehabilitation-requiring COPD increases and thus places increased demands on best possible use of the limited resources in the health care system, not least in the years to come [3].

Pulmonary rehabilitation (PR) is one of the cornerstones of care for patients with COPD. There is robust evidence that PR improves exercise capacity, enhances health-related quality of life (QoL) and reduces healthcare utilization [4–6]. PR is, therefore, strongly recommended in guidelines for COPD management [4, 7]. More than 85% of European countries, including Denmark, have implemented an outpatient model, where participants attend two to three sessions each week of supervised exercise and self-management training for a period of eight weeks or more [8]. Despite the compelling evidence for its benefits, PR is delivered to less than 30% of the patients with COPD likely to benefit from the intervention [9–13]. Referral and access to PR, however, remains a challenge for those with the most progressed stages of the disease [12, 14, 15].

A recently completed randomized clinical trial showed that approximately 1,100 patients with COPD annually were offered conventional hospital- and community-based PR during routine consultations in the Capital Region of Denmark, but of these, at least 700 patients (64%) declined participation [16]. Frequent transport to the center-based program, in the setting of distressing dyspnea and mobility limitation, is regularly reported as an access-barrier to attendance in hospitals and municipality health-care centers, and yet the delivery of PR has not changed significantly in the last 30 years [11, 17, 18].

Holland and colleagues addressed these barriers by investigating equivalence of a home-based PR and center-based PR in patients with COPD [19]. The results confirmed equivalence between home-based and center-based PR on short-term (eight weeks) improvements in

Inventory; PICOT, P: Population/Patient/Problem, I: Intervention, C: Comparison, O: Outcome and T: Time/Type of Study or Question; pcs., Pieces; PA, Physical activity; PR, Pulmonary rehabilitation; PTR, Pulmonary tele-rehabilitation; QALY, Quality-adjusted-life-years; QoL, Quality of life; RCT, Randomized controlled trial; RM, Repetition of maximum; REPORT, Rethink pulmonary rehabilitation; sec., Seconds; SPPB, Short Physical Performance Battery; SPIRIT, Standard Protocol Items: Recommendations for intervention Trials; TIDieR, Template for interventions Description and Replication; BPI, The Brief Pain Inventory; EQ-5D-3L, The EuroQol-5D-3L; 1-min-STS, 1-minute sit to stand; 30-sec-STS, 30-second sit to stand.

functional capacity and health-related QoL. However, the achieved gains in both arms were not maintained 12 months from baseline/end of intervention. More recently Zanaboni and colleagues investigated long-term (two years) effect from both supervised telerehabilitation and unsupervised treadmill training at home compared to standard care (i.e. no intervention) on incidence rate of hospitalizations and emergency department visits [20]. Their results indicated that the intervention groups experienced a better health status versus control group after one year and that home-based rehabilitation is feasible and might facilitate a higher attendance rate for PR.

The existing evidence shows that homebased rehabilitation and tele-rehabilitation are feasible and safe, however much is unknown regarding patients with COPD who deem themselves unable to participate in conventional PR programs as no studies, to our knowledge, have been conducted to specifically intervene in this group. Important knowledge in terms of symptoms (respiratory, anxiety, depression, and bodily pain), QoL, sleep quality, physical functioning, and response to novel home-based rehabilitation models are a black box with no substantial evidence, except for studies on qualitative perceived enablers and barriers for attendance [10, 11, 17]. Thus, we have a key gap in evidence for new rehabilitation models for patients with COPD unable to attend center-based programs including how it appeals to needs and barriers. The effectiveness and feasibility of pulmonary tele-rehabilitation (PTR) and home-based pulmonary rehabilitation (HPR) have never been investigated in this specific group labelled "center-based non-participants" (decliners) of patients with COPD. Yet, this is a strongly endorsed research scope by the American Thoracic Society/European Respiratory Society and more recently in their official clinical guidelines from May 2023, that states a strong recommendation for offering tele-rehabilitation for patients with COPD [9, 21].

PTR and HPR [14, 19] are two alternative models using health-care supportive technology that have proven equivalent or non-inferior to the conventional PR programs in patients with COPD who are able and willing to participate in conventional PR [9, 22]. However, for PTR and HPR to fulfil their potential, they must be accessible and well accepted by the target group of patients with COPD *unable* to attend the center-based programs, deliver the essential components of PR and be easy to implement. Furthermore, the benefits after the intervention must be of clinical relevance, i.e. superior to the current 'usual care' (medication and scheduled follow-up visits). Specific tools and delivery models that actively include and help this specific group of patients to improve and subsequently maintain their symptom control and functional level are needed [9, 14, 19–24]. This study–the REPORT study–investigates new models to increase the outreach to "center-based non-participants". The choice of the inclusion of control group is taken as no PR (usual care) is provided as standard treatment if the patient unable to participate in center-based programs.

The aims of this randomized controlled study are to (1) compare the clinical benefits of the two rehabilitation models with 'usual care', and (2) investigate adherence to and effect of an exercise maintenance-program (secondary endpoints).

## Objectives

**Primary research objective and question (PICOT).**   Is 10-weeks of group supervised PTR and individualized HPR equivalent to each other, and are the two interventions superior to usual care (no PR) on improving respiratory symptoms measured using the COPD Assessment Test (CAT) at 10-weeks from baseline in patients with COPD declining participation in center-based PR programs?

Population: Patients with COPD and moderate to severe airflow obstruction unable to access conventional center-based PR

**I**ntervention: 10-week group supervised PTR or 10-week HPR or **C**ontrol: Usual care (no intervention, besides routine and acute medical visits)

**O**utcome: Patient-reported respiratory symptoms (CAT score) (primary outcome)

**T**ime frame: The primary endpoint is 10-weeks from baseline

**Secondary research objective and question (PICOT).** Is 65-weeks of group supervised maintenance-telerehabilitation superior to usual care (no maintenance) on change/maintenance in respiratory symptoms at 75-weeks from baseline, in patients with COPD declining participation in center-based PR programs?

**P**opulation: Patients completing 10-weeks of group supervised PTR and/or individualized HPR

**I**ntervention: 65-weeks of group supervised telerehabilitation once weekly for 60 min.

**C**ontrol: Usual care (no intervention, besides routine and acute medical visits) and patient opting from 65-weeks maintenance program

**O**utcome: Patient-reported respiratory symptoms (CAT score) (primary outcome)

**T**ime frame: The secondary endpoint is 75-weeks from baseline

**Hypotheses.** Our hypotheses in a three-arm randomized controlled trial (RCT) design are

1. Effect of PTR and HPR will be equivalent to each other on respiratory symptoms (CAT score) at the primary endpoint (10-weeks)

2. Effect of PTR and HPR will be superior to usual care, that is the control group (CON—no intervention) on respiratory symptoms (CAT score) at the primary endpoint (10-weeks)

After completion of intervention in the PTR and HPR group, we consider a between-group difference of 2.5 points or larger (compared to usual care) in the CAT score [25] as a clinically relevant difference (MID) at 10-weeks (primary endpoint). Bounds for the equivalence limits between the interventions PTR and HPR are preset as difference within ± 2.5 for CAT score difference.

For the secondary endpoint (65-weeks maintenance-telerehabilitation) and intermediate measurement (35-weeks), we will also consider a between-group difference of 2.5 points (compared to usual care) in CAT score as a clinically relevant difference.

## Materials and methods

### Study principles

This protocol follows the Standard Protocol Items: Recommendations for intervention Trials [26] (SPIRIT) 2013, and the Template for interventions Description and Replication (TIDieR) checklist [27] for description of the intervention. After completion, the reporting of study findings will follow the Consolidated Standards of Reporting Trials (CONSORT) statement for non-pharmacological trials [28]. Outline of the study components as per SPIRIT checklist are shown in (Fig 1).

### Study design

The REPORT study is a prospective assessor and statistician blinded, three-arm, multicenter RCT. Assessor- and statistician are blinded to the group allocation. Block randomization will be applied. All procedures are approved by the Regional Ethical Committee of the Capital

| | Enrolment | Allocation | Post-allocation | | | | Close-out |
|---|---|---|---|---|---|---|---|
| **TIMEPOINT****| **-$t_1$** | **0** (After baseline test) | **$t_1$** Baseline | **$t_2$** Week 10 | **$t_3$** Week 35 | **$t_4$** Week 75 | **$t_x$** |
| **ENROLMENT:** | | | | | | | |
| Eligibility screen | X | | | | | | |
| Informed consent | X | | | | | | |
| Allocation | | X | | | | | |
| **INTERVENTIONS:** | | | | | | | |
| *PTR – 10 weeks* | | | ◆———————◆ | | | | |
| *HPR – 10 weeks* | | | ◆———————◆ | | | | |
| *Control – 10 weeks* | | | ◆————————————————◆ | | | | |
| *Maintenance – 65 weeks* | | | | | ◆————————◆ | | |
| **ASSESSMENTS:** | | | | | | | |
| **Primary outcome:** COPD Assessment Test (CAT) | | | X | X | X | X | |
| **Secondary outcomes:** Hospital Anxiety and Depression Scale (HADS), EuroQoL-5D-3L (EQ-5D-3L), Brief Pain Inventory (BPI), Multidimensional Fatigue Inventory (MFI-20), Pittsburg Sleep Quality Index (PSQI) | | | X | X | X | X | |
| *Physical activity and function* ActivePAL triaxial accelerometer (PAL), 1-minute sit to stand (1-min-STS), 30-second sit to stand (30-sec-STS), Guralnic test, 3m gait speed, 5-times sit-to-stand (SPPB), Hand-grip strength (Jamar dynamometer) | | | X | X | X | X | |
| Adherence to intervention/self-maintenance | | | | X | X | X | |
| Adverse events recorded | | | | X | X | X | |
| Number of hospital admission | | | past 12-mo | X | X | X | |
| Length of stay – hospital admission | | | past 12-mo | X | X | X | |
| Consultant visits | | | past 12-mo | X | X | X | |
| Mortality | | | | X | X | X | |
| Comorbidities | | | X | X | X | X | |
| **Descriptive variables:** FEV1/FVC ratio in % | | | X | | | X | |
| FEV1 (% predicted) | | | X | | | X | |
| DLCO (% predicted) | | | if available | | | if available | |
| Body Mass Index (BMI) | | | X | X | X | X | |
| Body weight (kg) | | | X | X | X | X | |
| Body height (cm) | | | X | X | X | X | |
| Fat Free Mass Index (FFMI) | | | if available | if available | if available | if available | |
| Smoking status | | | X | X | X | X | |
| Medication prescribed | | | X | X | X | X | |
| Bone fractures | | | past 12-mo | X | X | X | |
| BODS | | | X | | | X | |
| Charlson's Comorbidity Index (CCI) | | | X | | | X | |

**Fig 1. Outline of study as per SPIRIT checklist.**

Region of Denmark. The trial is registered at ClinicalTrials.gov–identification number: NCT05664945.

## Study setting

The trial is conducted as a multicenter study at the Respiratory Departments of seven hospitals in the Capital Region of Denmark. The following hospitals participate in the RCT: Bispebjerg-Frederiksberg, Herlev-Gentofte, Glostrup (respiratory department closed December 2023), Frederikssund-Hilleroed, and Amager-Hvidovre University Hospitals. A flow diagram of study design is presented in (Fig 2).

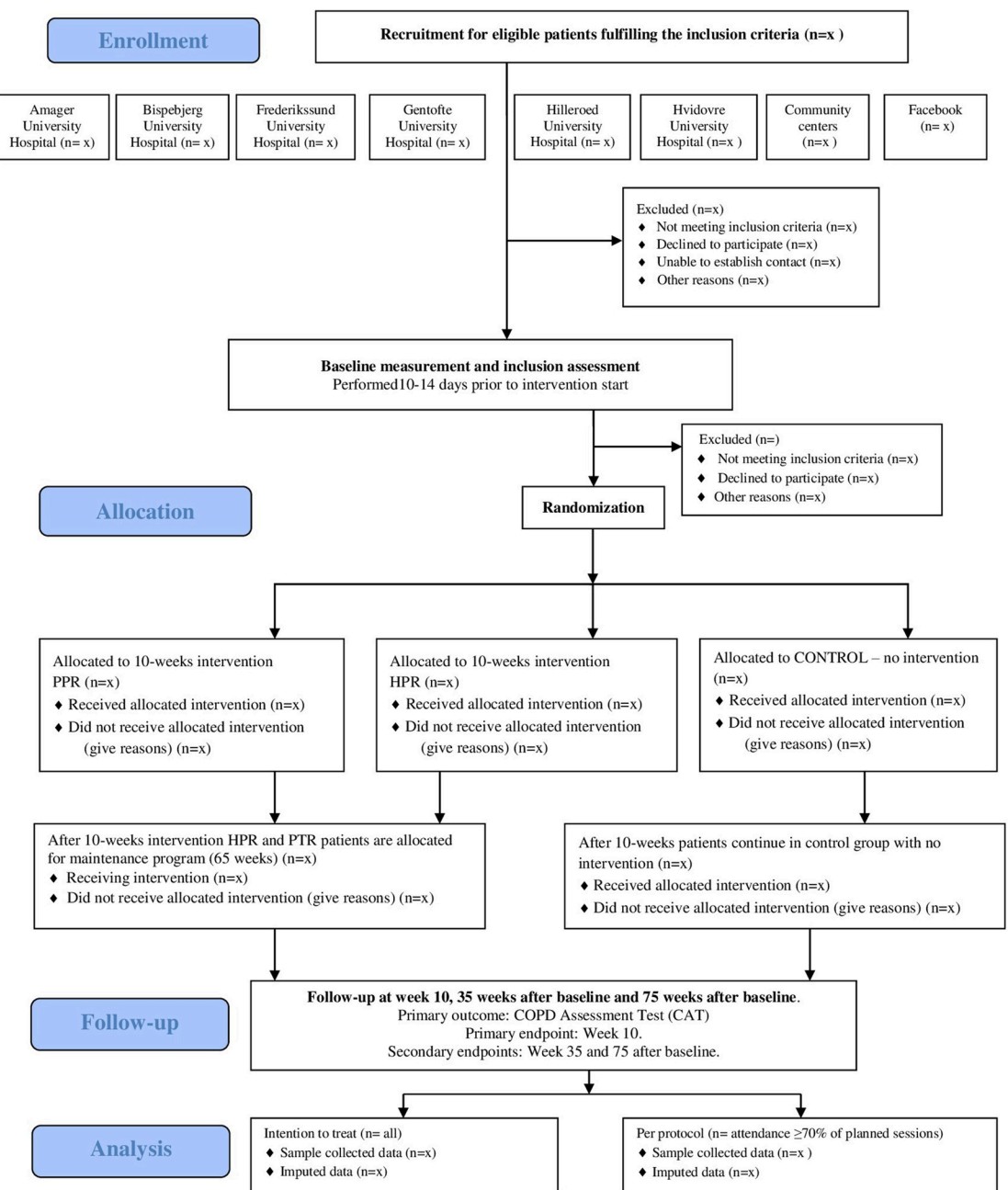

**Fig 2. Consolidate Standards of Reporting Trials CONSORT flow diagram of trial design.**

## Participants

The study aims at recruiting 180 patients with COPD. Recruitment and data collection of eligible patients started January 10[th] of 2023 and is estimated to continue to June 30[th] of 2025 but will continue until the preset sample size is reached. Participating sites provide information on patients who accept or decline participation. Recruitment is facilitated by local monthly meetings with clinical staff who pre-screen and recruit patients with the purpose of keeping attention on the progress of the study. Furthermore, a steering committee has been formed with

participation from representatives from all participating hospitals and is scheduled to meet twice a year.

## Eligibility criteria

**Initial contact.** Respiratory nurses at out-patient clinics identify potentially eligible patients with COPD at mandatory/scheduled control visits and send information to investigator (CN) who thereafter contacts the patients. In the investigators initial contact to determine patient eligibility to enrollment, reasons for not attending or resigning from conventional PR is explored and documented in the study database. Patients who decline attending the project is likewise documented with reasons in the study database. In cases of doubt concerning patient eligibility, for instance cognitive impairment or comorbid conditions, the projects pulmonologist and primary investigator is sough for advice and a pre-meeting is planned, whereafter the final decision is made. This in acceptance from the patient.

**Inclusion- and exclusion criteria.** *Inclusion criteria.*

1. Indication for PR according to national guidelines

2. Unable to access and participate in the conventional out-patient hospital- or community-based PR when offered during routine consultation

3. A *post-bronchodilator* ratio FEV1/FVC <70% (confirmed physician diagnosis of COPD) [7]

4. *A post-bronchodilator* FEV1 <80% (degree of airway obstruction) corresponding to GOLD grade 2–4 (moderate to very severe) [7]

5. GOLD groups B, C, D (GOLD 2023 update; GOLD E (former groups C and D)) [29] corresponding to severe respiratory symptoms and/or frequent acute exacerbations

6. Able to stand up from a chair (height 44-46cm) and walk 10 meters independently (with or without a walking aid)

7. Able to lift both arms to a horizontal level with a minimum of 1 kilograms dumbbells in each hand

*Exclusion criteria.*

1. Participation in conventional PR in the past 24 months

2. Cognitive impairment–unable to follow instructions

3. Impaired hearing or vision–unable to see or hear instruction from a tablet

4. Unable to understand and speak Danish

5. Comorbidities where the exercise content is contraindicated (including treatment for diabetic foot ulcer, active cancer treatment, life expectancy <12-months)

All eligible patients receive written information from the nurses about the study; a short version (pamphlet) and a longer version (approved by the Regional Committee of the Capital Region of Denmark). The investigator or project staff hereafter inform the patients verbally about the project. To ensure that all questions or concerns that patients might have, the investigator calls the patient to provide further information before the patient are asked to participate. All patients are encouraged to consider their consent for at least 24 hours before their final decision; this is according to Danish ethical guidelines for medical research. Patients who consent to participate in the study receives a written consent form, which they are asked to fill

in upon inclusion in the study. The patient will keep the original consent form and a copy will be archived with the Case Report Form (CRF).

**Ethics and consent to participate.** The trial protocol is approved by the Ethics Committee of The Capital Region of Denmark (H-22015777; 29.08.2022), by the Danish Data Protection Agency (P-2022-245-13101, 25.05.2022) and the trial is registered at ClinicalTrials.gov, identifier: NCT05664945 (23.12.2022).

The Ethics Committee will be informed about important protocol modifications for approval and if unexpected adverse advents should occur as mandatory by law. The study will be performed in accordance with the Declaration of Helsinki. A completed patient informed consent form is required from all patients participating in the study and must be signed by the patient and the informing physiotherapist or respiratory nurse. All investigators obtaining consent are qualified and appropriately trained.

## Baseline assessment

After identification of an eligible patient, the patient will be assigned a pseudo number as identification throughout the study. Hereafter the investigator or project staff schedules the patient for baseline assessment which will be conducted no earlier than 10–14 days prior to an eventual intervention onset. The assessment is performed preference specific in relation to whether the patient wants to be tested at their associated hospital or in their own home. The duration of the assessment is approximately 1.5 hours.

## Randomization

After obtained written informed consent and completion of baseline assessment the patient will be randomly allocated to one of the three following groups: PTR, HPR or CON (control, i.e. usual care). Randomization will be performed using a computer-generated block randomization list. The block size is n = 18; made by a biostatistician (TK) and PI (HH). A block will in randomly order consist of 7:7:4 allocation ratio (PTR, HTR and control, respectively). The randomization list is not accessible to the investigators, statistician or project staff involved in the conduct of the study.

## Blinding

To ensure concealment of allocation, a senior manager at another research project and with no interest or role in the project, will perform the randomization procedure, keep the master file and thus be responsible for the randomization. The senior manager will inform the investigator or project staff about the allocation, whereafter they will inform the patient about the allocation by phone call. After allocation the patient are considered included to either PTR, HPR or CON. The intervention will be scheduled to begin 10–14 days after randomization/baseline assessment.

All assessors are blinded to group allocation and previous test results. As the study is an intervention study the patients cannot be blinded, but prior to the assessments they are reminded on phone and SMS not to disclose their group allocation to the assessors. Regular fidelity checks of assessors will be conducted. The biostatistician who will perform data analyses and validated the results will be blinded to group allocation; this is to avoid the investigators subconscious bias. The research group will interpret the results, and the conclusion will be prepared in three versions before the allocation code is broken (one assuming that arm A is the intervention, one assuming that arm B is the intervention and one assuming that arm C in the intervention).

## Sample size

Three primary pairwise comparisons are made in the study, equivalence test between the two interventions and the superiority test for each intervention compared to the control group for the primary outcome of respiratory symptoms (COPD assessment test). Sample size estimation for the equivalence is based on a t-test with equivalence margin of -2.5 to 2.5 (clinically relevant difference), standard deviation (SD) of 4.5, power of 80% and significance level of 5%, and estimated 56 patients necessary in each group. For both, superiority tests a minimal clinical difference of 2.5, SD of 4.5, power of 80% and significance level of 5% were used in a two-sample t-test, with fixed sample of 56 patients the intervention groups. This yields a necessary size of 32 patients in the control group. The sample size is increased to accommodate an expected withdrawal of 20% in each group and thus a total of 180 (70/70/40) patients.

## Intervention groups

**Healthcare professionals.**   The PTR and HPR interventions are supervised and performed by skilled physiotherapists and respiratory nurses who have at least two years of experience with PR. The delivery of the interventions is ensured from the project group located at Hvidovre Hospital, Respiratory Department, Denmark.

**Equipment.**   The Capital Region of Denmark will provide and deliver all equipment and full day-to-day service.

**Training equipment.**   All patients in the intervention groups receive a tablet, dumbbells 1–5 kg. with the possibility of progression to 6–20 kg. or higher if needed and a two-level adjustable step box.

**Technology equipment (hardware and software).**   All patients in the intervention groups are provided an android tablet with a conference system and internet installed. The android tablet is a Samsung touch screen, size 10.1 inch, with a single interface that can be operated by all patients. The tablet ensures that the patients can communicate, visually and verbally, via a virtual platform (conference system: Pexip Inifinity Connect, version 1.10.3) with the project staff at Hvidovre Hospital and the remaining group participants. The screen for healthcare professionals is a 50.0 inch widescreen with access to a professional video conferencing system (Pexip Infinity Connect, version 1.10.3) that makes it possible to see multiple patients during a training session and train the group synchronously in real time. Furthermore, the system enables the possibility to show presentations and videos in patient education. The group TELE-sessions are delivered from Hvidovre Hospital, a room located in the Pulmonary Outpatient Clinic. The individual sessions are delivered from a room located at Hvidovre Hospital, Bispebjerg Hospital or an office where anonymity is possible.

The system is approved by Center for IT and Medico-Technology (CIMT) and the data protection Agency, The Capital Region of Denmark. The product supports the possibility of group training via video conference and is already used in other telemedical consultations.

**Prior experience with the technology.**   The research staff has in-dept knowledge and experience with the technology. From our previous telerehabilitation study, we documented, that technical issues leading to cancellation and/or rescheduling of group sessions affected only 2 of 360 group sessions [14]. Minor temporary technical issues (that is sound artefacts, screen freezes) not leading to cancellation or delay were present in 14% of the total group session (49 of 360). Individual patient cancellation caused by technical problems occurred in 12 of 1902 individual connections [14].

**Warm-up and cool-down in PTR and HPR intervention groups.**   The purpose of warm-up in the intervention groups are to familiarize the exercise movements, ensure adequate range of motion, stimulation of joints, and muscles and cardiorespiratory warm-up according

to the recommendations of the American College of Sports Medicine [30]. Warm-up have a duration of 5 min. in each group. The purpose for cool-down is to promote recovery and return to body to pre-exercise level.

Each stretch will be held for 30 sec. in according with ACSM [30] targeting the pectoralis major, neck lateral flexors, hamstrings and triceps surae muscles. Cool-down has the duration of 5 min. in each group. Warm-up and cool-down protocol are presented in Table 1.

**Pulmonary tele-rehabilitation group.**   The PTR group will consist of 4–6 patients who exercise at home and communicate via tablet, that is a home-based procedure that allows patients to avoid transportation to/from center-based PR. Each session is 60 min., i.e., 35 min. of exercise and 25 min. of patient education, two times a week for 10-weeks. The specific exercises performed in the PTR intervention have been used in more randomized intervention studies on patients with COPD [14, 31] and follow the exact same protocol. The exercises involve the larger muscle groups targeting 50% exercises for upper extremities and 50% for lower extremities [6, 14, 19, 31]. The specified volume, intensity and content for the exercise protocol are in accordance with national and international exercise recommendations [4, 6]. As the patients are expected to improve their mental and physical capacity during the intervention period a relative progression every fourth week are included in the exercise sessions.

Additionally, to the supervised program, patients are instructed to perform daily physical activity of at least 30 min. by modalities accessible to the patient, e.g. in- or outdoor walking activities in bouts of minimum 10 min.

The exercise program consists of six exercises and are presented in Table 2. The exercises are executed in four sets consecutive to achieve peripheral muscle failure and secondary dyspnea. Each set is carried out in a predefined period; week 1–2: 20 sec. active period and 40 sec. pause, week 3–6: 30 sec. active period and 30 sec. pause, week 7–10: 40 sec. active period and 20 sec. pause (Table 2). The goal for each set is to perform a maximum number of repetitions within the active period, i.e. 8 to 25 repetitions depending on patients exercise capacity and motivation, but with the aim of 12 to 20 repetitions. The exercise velocity is based on

**Table 1. Warm-up and cool-down protocol for PTR and HPR group.**

| Warm-up | Intensity and progression |
|---|---|
| • Heel uprisings<br>• Knee extension<br>• Rear deltoid row<br>• Chest press movement<br>• Vertical shoulder press<br>• Walking on site<br>• Side to side walking<br>• Leg curl<br>• Leg swing<br>• Squats<br><br>Duration: 5min. | Non-specific intensity<br><br>Purpose:<br>• increase body temperature<br>• cardiorespiratory warm-up<br>• muscle and tendon warm-up |
| **Cool-down** | **Intensity and progression** |
| • Breathing/relax exercises<br>• Pursed lip breathe<br>• Resting on a chair<br>• Relaxation exercise<br>• Intake water/ fruit/ other<br><br>Duration: 5 min. | Non-specific intensity, progression or choice of exercise |

Table 1 presents a protocol for exercises for warm-up and cool-down applied in both groups.

**Table 2. Exercise protocol PTR group.**

| Group supervised Pulmonary tele-rehabilitation | | | | | | | | |
|---|---|---|---|---|---|---|---|---|
| Exercise # | Exercise name | Extremities / function | Uni/ bilateral execution | Body position | Time/Volume | Exercise load | Intensity | Progression and work-period |
| 1 | Sit-to-stand from chair | Lower extremities | Bilateral | Sitting and standing | Active: 80–160 s. Rest: 160–80 s. Total: 240 s. | Bodyweight and dumbbells | Self-rated Borg CR-scale 10 – score 4–7 and/or reaching peripheral muscular failure. | Familiarization week 1–2: work of 20 sec. / rest 40 sec. |
| 2 | Biceps curl–shoulderpress | Upper extremities | Bilateral | Standing | Active: 80–160 s. Rest: 160–80 s. Total: 240 s. | Dumbbells | Repetitions: 8-25RM | Progression 1 week 3–6: work of 30 sec. / rest 30 sec. |
| 3 | Step-up on step-box | Lower extremities | Bilateral | Standing | Active: 80–160 s. Rest: 160–80 s. Total: 240 s. | Bodyweight, dumbbells and step-box | Sets: 4 sets of each exercise. | Progression 2 week 7–10: work of 40 sec. / rest 20 sec. |
| 4 | Bent over rowing | Upper extremities | Bilateral | Sitting Upper body slightly bend forward | Active: 80–160 s. Rest: 160–80 s. Total: 240 s. | Dumbbells | Load: body weight + 1 to 20kg external weight Rest period between exercises vary from 30–120 sec. After #1 and #5 up to 120 sec. rest before next exercise. After #2, #3, #4 and #6 app. 30 to 60 sec. rest. | Progression with external weights individually adjusted |
| 5 | Sit-to-stand from chair | Lower extremities | Bilateral | Sitting and standing | Active: 80–160 s. Rest: 160–80 s. Total: 240 s. | Bodyweight and dumbbells | | |
| 6 | Front raise dumbbells | Upper extremities | Unilateral | Standing | Active: 80–160 s. Rest: 160–80 s. Total: 240 s. | Dumbbells | | |

Table 2 presents exercises, function, execution, body position, time/volume, exercise load, intensity, rest period and progression.

recommendations applying to high-repetitive exercises, (> 15 repetitions) [30], i.e. moderate to high speed equal to 1–2 sec. for both the concentric and eccentric phrases of the movement. The exercise load consists of body weight supplemented with external dumbbells (1–20 kg.). The intensity is estimated to be equivalent to 40–80% of one repetition maximum (8–25 repetitions) and exercises are performed as high repetitive time-based muscle endurance at least 80% of the exercise time corresponding to a weekly volume of 60 min. (30 min. x 2). In practice, patients assess their training intensity using the self-rated Borg CR-10 scale, with a range from 0–10, aiming to exercise at a level of score 4–7 (moderate to very strong shortness of breath during exercises) and/or reaching peripheral muscular failure. For each session, the number of completed sets and the weight of the dumbbells used for each exercise are documented in an exercise log for all exercises to track progression.

The initial 2 weeks are dedicated to serve as a familiarization phase with the purpose of adapting to exercising, adjusting and optimizing load and preventing musculoskeletal overload injuries.

Exercises for the lower extremities (Table 2: exercise # 1,3,5) are carried out without dumbbells at the first session. If a patient can perform three consecutive sets without requiring rest during the active period, external load is added at the subsequent training session. The external load increase ranges from 2–4 kg. (total weight for two dumbbells) when progression adjustments are made. Exercises for the upper extremities (Table 2: exercise # 2,4,6) are carried out with the lightest weights (1 kg. / pcs.) at the first session. If a patient can perform three consecutive sets without requiring rest during the active period external load is added at the following training session. The external load increase ranges from 2–4 kg. (total weight for two dumbbells) when progression adjustments are made. Progressions are evaluated on an individual basis from session to session [32–35]. Additionally, patients are asked to count the number

of repetitions in each set every 6[th] session. If the number of repetitions exceeds 25, the external load is increased at the following training session.

**Patient-education.** The education topics follows the national guidelines [4] and are conducted as a combination of dialogue, reflection exercises, and practical exercises. Overall, the topics are similar in the PTR and HPR groups, but in the PTR group it is delivered as 20 min. sessions two times a week in total of 20 sessions. In the HPR group it is delivered as appr. 15 min. sessions once a week in a total of 10 sessions. The primary focus is to promote smoke cessation, correct intake of medication and increasing physical activity/reducing sedentary behavior. PTR topics are presented in Table 3, HPR topics in Table 4.

**Home-based pulmonary rehabilitation group.** Patients allocated to the HPR group will participate in individualized self-initiated home-based PR. The patient exercises at home in sessions preferably three times a week at least for 20 min. during the 10-week intervention. Weekly, stepwise and motivational goal setting are used to reach the overall exercise goal. For more motivated patients exercising more than three times a week is welcomed to meet the individual desires.

The individualized exercise prescription will follow the Frequency-Intensity-Time-Type-Volume-Progression principles (FITT-VP) [30] combined with the patient's motivation and goals (please see Tables 5 and 6). Exercises are evidence-based and used in several intervention studies on patients with COPD and involves larger muscle groups exercises for upper/lower extremities [6, 14, 19]. All patients are recommended to start with a minimum of three exercises (Table 5, core exercises #1,2,3). Add-on exercises are determined based on patient motivation, wishes and goals. A quality check of exercises will be performed online twice during the 10-week intervention.

The first session is a visit held at the patient's home. During the visit the respiratory physiotherapist/nurse establishes exercise goals with the patient, writes a formal exercise prescription of 3–6 exercises, and educates the patient in the use of an exercise-log. The physiotherapist supervises the prescribed exercises and assists in filling out in the exercise-log. Secondly, the patient is encouraged to complete a mindfulness program of choice for 15 min. at least once a

**Table 3. Educational topics for PTR group.**

| Topic / themes | Communication / learning form | Duration | Number of sessions |
|---|---|---|---|
| Welcome and individual presentations | Information, dialogue | 20 min. | 2 |
| What is COPD and treatment available* | Information, dialogue | 20 min. | 1 |
| Medication intake/inhalation technique# | Information, dialogue | 20 min. | 1 |
| Respiratory techniques | Information, dialogue, reflection and practical exercises | 20 min. | 2 |
| Early signs of exacerbation and action plan | Information, dialogue and reflection | 20 min. | 1 |
| Smoking cessation and substitution | Information, dialogue and reflection | 20 min. | 1 |
| Mindfulness | Practical exercises | 20 min. | 3 |
| Physical activity and habits | Information, dialogue and reflection | 20 min. | 2 |
| Nutrition, importance of food in COPD | Information, dialogue, reflection and practical exercises | 20 min. | 1 |
| Sleep and fatigue | Information, dialogue and reflection | 20 min. | 1 |
| Pain, strategies to reduce pain in COPD | Information, dialogue and reflection | 20 min. | 1 |
| Repetition | | 20 min. | 2 |
| Group needs | | 20 min. | 2 |

Table 3 presents possible educational topics for PTR group. The topics follow the national guidelines.

* Topic delivered by pulmonologist.

# Topic delivered by respiratory nurse. The remaining topic are delivered by physiotherapist or respiratory nurse.

**Table 4. Educational topics for HPR group–based on individual needs.**

| Topic / themes | Communication / learning form | Location | Duration |
|---|---|---|---|
| Welcome and presentation of intervention | Information, dialogue | During first home visit | 15 min. |
| What is COPD and treatment available | Information, dialogue | Via tablet | 15 min. |
| Medication intake/inhalation technique | Information, dialogue | Via tablet | 15 min. |
| Respiratory techniques | Information, dialogue, reflection and practical exercises | Via tablet | 15 min. |
| Early signs of exacerbation and action plan | Information, dialogue and reflection | Via tablet | 15 min. |
| Smoking cessation and substitution | Information, dialogue and reflection | Via tablet | 15 min. |
| Mindfulness | Practical exercises | Via tablet | 15 min. |
| Physical activity and habits | Information, dialogue and reflection | Via tablet | 15 min. |
| Nutrition, importance of food in COPD | Information, dialogue, reflection and practical exercises | Via tablet | 15 min. |
| Sleep and fatigue | Information, dialogue and reflection | Via tablet | 15 min. |
| Pain, strategies to reduce pain in COPD | Information, dialogue and reflection | Via tablet | 15 min. |
| Other topics based on individual needs | | Via tablet | 15 min. |

Table 4 presents possible educational topics that can be chosen in HPR group. The individual patient can choose other topics based on their needs and desire. The duration and number of sessions can vary given the importance of the topic.

week if it appeals to the patient (optional). Furthermore, the patients are given a COPD educational booklet, a pamphlet on diet and COPD, medication etc.

Additionally, daily physical activity of at least 30 min. by modalities accessible to the patient, e.g. in- or outdoor walking activities in bouts of minimum 10 min. is recommended.

In the exercise-log the patient is instructed to record when, where and how often she/he plans to exercise, what might obstruct the plans, and what can be done to overcome any perceived barriers. The exercise log includes contact information to enable the patient to contact the physiotherapist/nurse between sessions if required. During the initial home visit the physiotherapist/nurse assesses safety of the home environment in relation to the exercise program.

The first home session is followed by nine sessions, one weekly, with structured motivational and professional counseling, via tablet or telephone (choice of patient's preference) by a physiotherapist/nurse for 10-weeks using a motivational interviewing approach. Day and time for the weekly call is agreed upon with the patient at their preference in mind on a weekly basis.

The weekly call will begin with a brief talk of what has happened during the exercise program over the past week (prior to the call the patient has sent a text message via phone with a picture of the last week exercise log). Consistent with the principles of motivational interviewing, the physiotherapist/nurse might then ask the patient why she/he might want to increase the exercise program (i.e. time and/or intensity) (desire); how she/he might do this if she/he decides to (ability); what would be the most important benefits in doing more exercise (reasons); and how important it is for the patient to perform more exercise at this time (need). The physiotherapist/nurse will encourage the patient to move towards commitment and action, with specific goals set for the following week (please see Table 6). During the call the patient document their goals in their diaries in the same manner as from the initial home visit session. The discussion will then move on to other issues in relation to COPD. The patient is provided with a written menu of topics relevant for COPD and self-care and encouraged to select a topic of relevance for discussion, providing opportunity for self-management. For a concise overview of the differences between PTR and HPR group, please see Table 7.

**Control group.** Patients randomized to the control group will receive usual care; medication, scheduled follow-up visits at the respiratory outpatient clinics or general practitioner.

**Table 5. Exercise protocol HPR group.**

| **Home-based pulmonary rehabilitation** | | | | | | |
|---|---|---|---|---|---|---|
| Exercise # | Exercise name | Extremities / function | Uni/bilateral execution | Body position | Exercise load | Intensity | Progression and work-period |
| **CORE EXERCISES (must perform)** | | | | | | Self-rated BORG CR-10 dyspnea 4–7 OR reaching peripheral muscular failure<br><br>Repetitions:<br>8-25RM<br><br>Sets:<br>Up to 4 sets of each exercise.<br>Load: body weight + 1 to 20kg external weight<br><br>Exercises are adjusted according to FITT principles–for details see Table 6. | Familiarization week 1–2:<br>Exercise 2–3 times a week<br><br>Progression 1 week 3–6:<br>Exercise 3–4 times a week<br><br>Progression 2 week 7–10:<br>4–5 times a week or daily<br><br>Progression with external weights individually adjusted |
| 1 | Sit-to-stand from chair | Lower extremities | Bilateral | Sitting and standing | Bodyweight and dumbbells | | |
| 2 | Biceps curl–shoulderpress | Upper extremities | Bilateral | Standing | Dumbbells | | |
| 3 | Step-up on step-box | Lower extremities | Bilateral | Standing | Bodyweight, dumbbells and step-box | | |
| **CHOICE (patient choice exercises)–upper extremities** | | | | | | | |
| 4 | Bent over rowing | Upper extremities | Bilateral | Sitting or standing Upper body slightly bend forward | Dumbbells | | |
| 5 | Front raise dumbbells | Upper extremities | Unilateral | Standing | Dumbbells | | |
| 6 | Side-ways raise dumbbells | Upper extremities | Bilateral | Standing | Dumbbells | | |
| 7 | Seated row dumbbells | Upper extremities | Bilateral | Sitting | Dumbbells | | |
| 8 | "Flyers" dumbbells | Upper extremities | Bilateral | Sitting or standing Upper body slightly bend backward | Dumbbells | | |
| 9 | Upright row dumbbells | Upper extremities | Bilateral | Sitting | Dumbbells | | |
| **CHOICE (patient choice exercises)–balance and mobility** | | | | | | | |
| 10 | Step over box | Lower extremities | Bilateral | Standing | Bodyweight, dumbbells and step-box | | |
| 11 | High knee-lift | Lower extremities | Bilateral | Standing | Bodyweight and dumbbells | | |
| 12 | Lunges forward | Lower extremities | Bilateral | Stepping forward equivalent to a "long step" forward | Bodyweight and dumbbells | | |
| 13 | Side-shift weightbearing | Balance | Bilateral | Shifting weightbearing from side to side | Bodyweight | | |
| 14 | Forward-backward weightbearing | Balance | Bilateral | Shifting weightbearing forward to backward | Bodyweight | | |
| 15 | 1-legged stance | Balance | Unilateral | Standing on one-leg with or without support | Bodyweight | | |

Table 5 presents exercises, function, execution, body position, exercise load, intensity, rest period and progression

Except for assessment visits baseline and 10-week visit, 35-week and 75-weeks visits no intervention is offered. If a patient changes his/her mind and wishes to participate in a conventional hospital- or community-based PR program, it will be granted as this is a highly recommended treatment (e.g. rehabilitation after hospital admission due to exacerbation).

**Maintenance program.** After completion of the 10-weeks intervention period, patients randomized to PTR or HPR are offered to continue for free in a 65-week maintenance program (MP), consisting of a supervised group tele-exercise once weekly for 60 min. (appr. 45 min. of exercise and 15 min. to group discussions when relevant).

**Table 6. Home-based pulmonary rehabilitation–overview.**

| Session 1 | |
|---|---|
| **Where** | Patient preference: Held at the patient's home or at the hospital. |
| **Purpose** | Establish exercise goals, prescribe formal exercise, education of usage of exercise log, review and testing of exercises. |

| Session 2–10 | |
|---|---|
| **Preparation** | Prior the online consultation the patient has sent a SMS with picture of the exercise log to the physiotherapist or nurse. |
| **Where** | Online consultation via tablet (using conference system; Pexip Infinity Connect) |
| **How**<br><br>**Next step** | The physiotherapist or nurse conduct a motivational conversation with focus on **DARN** principals:<br>**D**esire = Does the patient want to increase the exercise program?<br>**A**bility = How will the patient do this if he/she decides to do so?<br>**R**eason = What would be the most important reasons to do so?<br>**N**eed = How importation is it for the patient to perform more exercises at this point?<br><br>*If the patient desires to adjust the exercise program the FITT-VP principles are used, as described in the next section.* |

*Exercises are prescribed will be individualized with consideration of health status.*

| | Frequency (How often is exercise done each week) | Intensity (How hard is the exercise) | Time (How long time is the exercise duration) | Type (What is the mode of the exercise) | Volume (What is the total amount of exercise) | Progression (How is the program advanced) |
|---|---|---|---|---|---|---|
| **Mode 1** | Increase frequency from 2 to 3 times a week | Peripheral muscle failure OR BORG CR-10 dyspnea 4–7 Increase amount of repetitions: 8-25RM (40–80% of 1RM) OR increase external load | Increase time with 1–5 min. | Adding different exercises such as balance, coordination. | A gradual progression of total amount of exercise during.<br><br>Increase number of sets: from 2 to 3 or more. | A gradual progression of greater resistance, and/or more repetitions per set, and/or increasing frequency. |
| **Mode 2** | Increase frequency from 3 times to 4 times a week | Increase amount of repetitions: 8-25RM (40–80% of 1RM) OR increase external load | Increase time with 6–10 min. | Adding exercises as walking in- or outside. | | |
| **Mode 3** | Increase frequency to 4–5 times a week or daily | Increase amount of repetitions: 8-25RM (40–80% of 1RM) OR increase external load | Increase time with $\geq$ 10 min | Adding exercises such as cycling in- or outside. | | |

Table 6 presents an overview from the procures at all sessions, including progression applying FITT-VP principles.

The overall goal with the MP is to maintain or improve respiratory symptoms and physical functioning, other symptoms as fatigue, pain and sleep and affect behavior change positively (e.g., sustainable exercise habits, increase physical activity level, achieve knowledge on practical exercises) using minimal equipment.

During the 65-week MP patients attend one online session a week and are instructed and encouraged to perform similar online prescribed physical exercises at home. The weekly goal is for the patient to achieve at least 30 min. of muscle-endurance based exercises twice weekly. The patients are asked to count their physical exercises and activity level via a home diary. Furthermore, patients are encouraged to explore available mindfulness exercises via the tablet. The prescribed physical exercises are available on the tablet as an instructed exercise-film. Seven mindfulness exercises are available on the tablet for self-initiated mindfulness-guidance.

The MP consists of four themes during the entire 65-weeks (Table 8). The period for each theme is five weeks and are performed in the following order: a) "standardized program", b) "balance theme", c) "eccentric theme", and d) "partner theme".

**Table 7. Short overview of differences from PTR and HPR intervention groups.**

| | PTR | HPR |
|---|---|---|
| **Supervised or self-initiated** | Supervised | Individual self-initiated |
| **Group or individual** | Group-based (4–6 patients) | Individual |
| **Exercise location** | Home-based | Home-based |
| **Delivery of sessions** | Tablet with video-conference system (Twice a week) | Tablet with video-conference system or telephone (choice of patient's preference) (Once a week) |
| **Duration of the sessions (focus)** | 60 min. (35 min. of exercise and 25 min. of patient education) | 15 min. (motivational counselling and program adjustment) |
| **Exercises days** | Twice a week administered by the physiotherapist or respiratory nurse | Self-administered by the patient (Monday to Sunday) |
| **Exercise principles** | Resistance-endurance training | FITT-VP (for more info please see Table 6) |
| **Exercises–target groups** | Targeting larger muscle groups– 50/50% lower-extremity and upper-extremity (please see Table 2) | Targeting larger muscle groups– 50/50% lower-extremity and upper-extremity (please see Table 5) |
| **Education** | Structured via the PT–following national guidelines | Structured via the PT and the patient based on need and desire |
| **Equipment** | 1 tablet with video-conference system, a step-box and pairs of dumbbells weights from 1 to 5 kg. (heavier dumbbells available) | 1 tablet with video-conference system, a step-box and pairs of dumbbells weights from 1 to 5 kg. (heavier dumbbells available) |

Table 7. The table present a short overview of the differences in PTR and HPR intervention. For more information please see Tables 2, 5 and 6. FITT-VP: Frequency, intensity, time, type, volume and progression.

The rationale for a balance theme is that secondary manifestations are detected in patients with COPD in form of impaired balance and deficits in postural control which increases fear of falling and risk of failing [36, 37]. The balance exercises aim to improve balance, increase knowledge on fall prevention strategies and give insight into simple, feasible and modifiable exercises, that can be performed in a home setting and as add-on exercises. The rationale for an eccentric theme is that eccentric movement is a key element of mobility and task performance [38] and more evidence is emerging regarding COPD [39], such as studies investigating eccentric cycling and downhill walking. Eccentric contractions occur during slow movement / deceleration movement thereby stretching the myofilaments due to high external load [38, 39]. This movement usually demands lower oxygen uptake, reduced heart rate and blood pressure [39], more specifically it increases muscle strength. The purpose of applying an eccentric workload is to induce peripheral muscle fatigue rather than dyspnea, create periodization in the scheduled muscle training program, provide the participants insight and knowledge on a difference strategy to muscle training / performing exercises.

The fourth theme is the "partner" theme, and the rationale is to apply "team spirit" / "gamification" perspective. The purpose of this block is to achieve cohesion in the group, create a motivational environment and provide opportunities to competition at an individual level.

The standardized program consists of the same six exercises used in the PTR group and the purpose it to continue with known exercises and for the patients from HPR group to familiarize with the supervised exercises and intensity.

The balance theme consists of the first three standardized exercises (Table 8 #1,2,3) followed by exercises with focus on balance. The exercises are presented in Table 8 and are previously used in other studies with patients with COPD [37]. All exercises are performed on an

**Table 8. Maintenance program–exercise and theme overview.**

| | Warm-up (duration 5 min.) | Intensity and progression | Cool-down (duration 5 min.) |
|---|---|---|---|
| **Warm-up and cool-down exercises applied in all sessions.** | • Heel uprisings<br>• Knee extension<br>• Rear deltoid row<br>• Chest press movement<br>• Vertical shoulder press<br>• Walking on site<br>• Side to side walking<br>• Leg curl<br>• Leg swing<br>• Squats | Non-specific intensity<br><br>Purpose:<br>• increase body temperature<br>• cardiorespiratory warm-up<br>• muscle and tendon warm-up | **Individual non-specific rest**<br>  • Breathing/relax exercises<br>    • Pursed lip breathe<br>    • Resting on a chair<br>    • Use of rest-room<br>    • Intake water/ fruit/ other<br>Non-specific intensity, progression or choice of exercise |
| **THEME 1** | **Exercises (30 min.)** | **Intensity and progression** | **Period** |
| **Protocol exercises** | *High repetitive/time-based muscle-endurance training*<br>Performed in numeric order:<br>1. Sit-to-stand from chair<br>2. Biceps curl–shoulderpress<br>3. Step-up on step-box<br>4. Bent over rowing<br>5. Sit-to-stand from chair<br>6. Front raise dumbbells | Intensity<br>30–30 (active/pause) sec.~ 8-25RM. 4 sets of each exercise<br>Load: body weight + 1 to 20 kg external weight.<br>Progression with external weights individually adjusted. | 5 consecutive weeks |
| **THEME 2** | **CORE exercises (15 min.)** | **Intensity and progression** | **Period** |
| **Balance training** | *High repetitive/time-based muscle-endurance training*<br>Performed in numeric order:<br>1. Sit-to-stand from chair<br>2. Biceps curl–shoulderpress<br>3. Step-up on step-box<br><br>*Balance exercises performed in numeric order (app. 25 min.)*<br>**Stance exercises (static and dynamic)**–<br>**1)** Narrow stance—Stand without support—Eyes closed (time 20 s)—Reach beyond BOS—Throw and catch ball—Count backward<br>**2)** One-legged stance—One-leg stand (time 30 s)—Visual targets (turn and look)—Spell names with foot—Leg out to the side -Eyes closed<br>**Transition exercises**–<br>**1)** Sit to stand—Sit to stand and pick up objects from floor,<br>**2)** Sit on floor and stand up: with chair for descent and ascent, No chair for either descent or ascent, No chair—Timed (safely and quickly),<br>**3)** Stairs (in this study: step-box)—Tap 10x with arm support—Tap 10x any speed—Tap 10x fast speed—Tap and count backward by 2 s—Tap and arm lift<br>**Gait exercises**–<br>**1)** Tandem walk with light finger support -<br>Tandem walk no U/E support—Sideways walk—Backward walk<br>Tandem walk while—spelling words—Backward walk while naming<br>words starting with "w" (in Danish words with "s")<br>**2)** Walking in open space—Fast walking (. 6 m)—Change in speed<br>Quick direction change—Walk and look—Walk and count backward<br>Walk and recite months of the year<br>**Functional strength exercises**–<br>**1)** Lower leg—Toe raises (arm support) 2x10 reps—Toe raises (no arm support) 10 reps, 3-s holds—Heel raises (arm support) 2x10 reps—Heel raises (no arm support)– 2x10 reps—Walk on toes—Walk on heels | Intensity for CORE exercises:<br>30–30 (active/pause) sec.~ 8-25RM. 4 sets of each exercise.<br>Load: body weight + 1 to 20 kg external weight.<br>Progression with external weights individually adjusted.<br><br>Intensity for balance exercises:<br>Work individual up to 3min in each exercise.<br>Progression exercises:<br>1. Reducing amount of support<br>2. Open / closed eyes<br>3. Cognitive add-on exercises<br>4. Variation of speed and movement | 5 consecutive weeks |

*(Continued)*

**Table 8.** (Continued)

| | Warm-up (duration 5 min.) | Intensity and progression | Cool-down (duration 5 min.) |
|---|---|---|---|
| **THEME 3** | **Exercises (30 min.)** | **Intensity and progression** | **Period** |
| **Eccentric training** | *Fast concentric–slow eccentric based muscle training*<br>Performed in numeric order:<br>1. Sit-to-stand from chair<br>2. Biceps curl–shoulderpress<br>3. Step-up on step-box<br>4. Bent over rowing<br>5. Sit-to-stand from chair<br>6. Front raise dumbbells | Intensity for eccentric exercises:<br>50/30 sec. (active/pause) 5–8 RM. 4 sets of each exercise.<br>Explosive concentric/acceleration upon onset of exercise (1–2 sec.) and eccentric/deceleration upon offset (4–8 sec).<br>Load: body weight + 1 to 20 kg external weight.<br>Progression with external weights individually adjusted.<br>Progression with time and external weights individually adjusted. | 5 consecutive weeks |
| **THEME 4** | **Exercises (30 min.)** | **Intensity and progression** | **Period** |
| **"Partner exercises"** | Performed in numeric order:<br>1. Sit-to-stand from chair<br>2. Biceps curl–shoulderpress<br>3. Step-up on step-box<br>4. Bent over rowing<br>5. Sit-to-stand from chair<br>6. Front raise dumbbells | Intensity for partner exercises:<br>To perform 8-25RM. 4 sets of each exercise or to muscle failure.<br>Load: body weight + 1 to 20 kg external weight.<br>Progression with external weights individually adjusted. | 5 consecutive weeks |
| **REST periods between exercises** | | | |
| | Rest period between exercises vary from 30–120 sec. After #1 and #5 up to 120 sec. rest before next exercise.<br>After #2, #3, #4 and #6 app. 30 to 60 sec. rest. | | |

Table 8 presents an overview of maintenance program with themes, exercises, intensity, progression and period.

individual level regarding balance impairments and need of support. The exercises consist of four types: 1) stance exercises, 2) transition exercises, 3) gait exercises, and 4) functional strength exercises. All patients start with static exercises and are progressed after individualized to more challenging means such as eyes closed and cognitive tasks.

The eccentric theme consists of the six standardized exercises (Table 8, #1–6) applying an eccentric focus while performing the exercises. The exercises are performed in the same manner as the standard protocol regarding set and external load. For time-period please see Table 8.

Patients are instructed to perform their exercises individually with focus on deceleration of movement, e.g. 4–8 sec. deceleration and explosive concentric acceleration e.g. 1–2 sec. All patients perform the exercises at their individual level of physical function.

The fourth theme is "partner" theme and the exercises are performed in the same manner as the standard protocol regarding set and external load, however the intensity is defined by the following: 1) team up with a partner (2 and 2) one patient performs as many repetitions as possible (to achieve muscle fatigue) and then the other patient takes over, 2) "Roll the dice"– the number of eyes on the dices (up to four dices) accumulate number of repetitions or the number of eyes on one dice represents the amount of seconds for the set, 3) "Team knowledge" with a partner vs. another team; test of knowledge where the team who gets to correct answer have a break, whereas the other team perform 10 repetitions of a given exercise.

The exercises are presented in Table 8. All exercises will be progressed or regressed, i.e. if a patient are able to perform the exercises with correct intensity and safety then the exercise can be progressed or in case of the opposite then the exercise can be regressed for instance

performing balance with more support. During standard protocol and eccentric theme external load is sought increased.

**Statistical analyses plan.** Analyses of the outcomes will be performed by the trial statistician who is blinded to the treatment allocations. Secondary analyses will be performed by the investigator (CN).

Baseline data will be reported as means and standard deviations (medians and interquartile ranges) or frequencies and proportions as appropriate. Data analyses of primary and secondary endpoints will be performed using intention-to-treat analyses. Additionally, per-protocol analyses including patients who complete ≥70% of the PTR and HPR rehabilitation program will be performed. Reporting will follow the CONSORT (Consolidated Standards of Reporting Trials) Statement for non-pharmacologic trials [28]. Differences between the groups' changes of primary and secondary outcomes (10-weeks from baseline; 35-weeks from baseline and 75-weeks from baseline) will be analyzed by mixed effect models. The models include adjustment for treatment groups, age, sex, BMI, $FEV_1$, Charlson Comorbidity Index and smoking status, and a random effect recruitment site with unstructured covariance matrix. To account for possible regression to the mean effect, the baseline measure for the outcome will be included as a fixed effect variable in the models. Normal distribution of the model residuals is evaluated by inspection of Q-Q plots. All data is expected to be missing at random, and missing values included in the analysis will be imputed by multiple imputation. Specific variables used for imputation will be explained and reported. If only a couple of values are missing sensitivity analysis replacing missing values with extremes may be used instead of multiple imputation.

Survival is visualized by Kaplan-Meier curves. Group differences on proportions of patients being adherent, hospitalization and death are analyzed using chi-squared test. No interim analysis will be performed. Statistical analyses are carried out using R 4.1.2. (R Foundation for Statistical Computing, Vienna, Austria). P-values of less than 0.05 are considered statistically significant.

**Compliance.** In addition to the intention-to-treat analysis, we will also perform a per-protocol analysis. The participants in both intervention groups must have completed 70% per cent of the scheduled exercise program to be included in the per-protocol analysis.

**Health economic analysis.** In a separate paper we will conduct cost-utility analysis (CUA) of the two new delivery models for PR delivered in a multi-center three-arm randomized controlled trial. The economic evaluation is conducted alongside the RCT calculating incremental cost-effectiveness ration. This CUA will followed international guidelines for the conduction of a CUA alongside a clinical randomized controlled trial [40, 41]. The EQ-5D-3L collected utility score will be used in the estimation of quality-adjusted-life-years (QALY).

**Data collection.** All data are collected in CRF's to uphold stringent data collection. Blinded assessors from three different hospitals (Bispebjerg, Hillerod, and Hvidovre) perform pre-, post- and follow-up test. All hospitals have two blinded assessors available who cover the entire Capital Region. The assessors have clinical experience and practice with patient-reported outcomes in questionnaires and physical performance tests. To ensure that all assessors follow the same test protocol, use standardized instructions and recordings of results, the assessors have completed an assessment calibration course. Furthermore, they have all been observed and evaluated performing live test on patients prior to start as blinded tester. To the extent possible, the same assessor tests a participant at all test times.

**Data management.** All CRFs and paper versions of questionnaires will be reviewed and checked for errors and missing data before entered in a secure Research Electronic Data Capture (Redcap, version nr. LTS 13.7.14) approved by Capital Region. All data will be double-checked prior to exporting it to relevant statistical software programs (R, IBM SPSS version 29

and GraphPad Prison version 10.1.2). The principal investigator will have full access to the entire dataset. Co-investigators and steering-committee will have access as needed for random auditing. All CRFs and questionnaires will be anonymized. To ensure confidentiality all CRFs will be placed in a secured and locked filling cabinet where only the principal investigator and co-investigator have access to. Data management will comply with the rules of the Danish Data Protection Agency.

**Outcomes.** All outcomes and time of measurements are presented in Table 9. All follow-up test will be conducted by a blinded assessor.

**Primary outcome measure.** *The COPD Assessment Test (CAT)* will be used to measure change in respiratory symptoms. The CAT is a patient completed questionnaire consisting of 8 items on self-reported health-status and symptom [42]. Each item is scored ranging from 0 to 5 points, where 0 indicate no impact or symptoms and 5 representing worst score. A total score range from 0–40 points. CAT is a validated questionnaire with a Cronbach's α of 0.88 and found responsive to change in self-reported health status and symptoms after pulmonary rehabilitation [25, 43]. A minimal clinical difference (MCID) of -2 to -3 points has been suggested [25]. For this study design we decided a MCID of -2.5 point will be of clinical relevance.

## Secondary outcome measures

**Questionnaires outcomes.** *The Hospital Anxiety and Depression Scale (HADS)* is questionnaire consisting of 14-items assessing anxiety and depression level in medically ill patients [44]. HADS consists of two sub scores; HADS anxiety (HADS-A) and HADS depression (HADS-D) comprising seven questions assessing anxiety and seven questions assessing depression, respectively. Each question ranges a score from 0–3 (where 0 = no symptoms). HADS has been validated with a Cronbach's α of 0.83 (HADS-A) and 0.82 (HADS-D) [44]. Scoring from 0–7 on each of the two sub-scales are considered within normal range, whereas scores of 8–10 suggest a potential risk of anxiety and/or depression disorder. Scores of 11 and above suggest the probable presence of anxiety and/or depression disorder [44]. 1.5 point has been suggested as MCID on both [45]. Smid et al suggested MCID ranging between -2.0 and -1.1 points for HADS-A and -1.8 and -1.4 or HADS-D [25]. Nikolovski et al suggested a MCID of $\geq$ 1.5 in HADS-A, HADS-B and HADS-T [46]. In this study -1.5 point on both scales will be used as MCID.

*The EuroQol-5D-3L (EQ-5D-3L)* is a generic self-reported global measure for health-related QoL. EQ-5D consists of 5 domains (mobility, self-care, usual activities, pain/discomfort, and anxiety/depression) and a 20 cm visual analog scale (EQ-VAS) ranging from zero (worst imaginable health) to 100 (best imaginable health). Each item in EQ-5D is scored ranging from 1–3 with 1 indicating no symptoms, producing a utility index range from -0.624 to 1.000. MCID of 6.5 to 8-points in EQ-VAS has been suggested [47]. Bae et al suggested a pooled MCID estimate of 0.028 for EQ-5D utility index [48]. We will use 6.5 point as MCID threshold for EQ-5D-VAS [49].

*The Brief Pain Inventory (BPI)* will be used to assess pain within a period of seven days [50]. The BPI consists of a "pain drawing" with indication of pain location and eight questions assessing pain on a visual analog scale ranging from 0–10 with 10 representing worst pain. A Cronbach α ranging from 0.91 (BPI magnitude domain) to 0.94 (BPI inference domain) has been suggested [51] The following cut-off scores has been suggested to assess severity of pain; mild pain (0–4), moderate pain (5–6) and severe pain (7–10) [52].

*Multidimensional Fatigue Inventory (MFI-20)* will be used to assess perceived fatigue [53]. The MFI-20 consists of 20 items with 5 domains (general fatigue, physical fatigue, mental fatigue, reduced motivation, and reduced activity). Each domain has four questions with a

**Table 9. Outcomes and flowchart of study measurements.**

| Outcomes | Variables | Visit 0 Baseline | Visit 1 End of intervention | Visit 2 Maintenance | Visit 3 Maintenance |
|---|---|---|---|---|---|
| | | Baseline | Week 10 | Week 35 | Week 75 |
| **Primary outcome** | | | | | |
| Respiratory symptoms | COPD Assessment Test (CAT) | X | X | X | X |
| **Secondary outcomes** | | | | | |
| *Self-reported health questionnaires* | | | | | |
| Anxiety and depression symptoms | Hospital Anxiety and Depression Scale (HADS) | X | X | X | X |
| Quality of life–generic | EuroQoL-5D-3L (EQ-5D-3L) | X | X | X | X |
| Body pain- intensity and location | Brief Pain Inventory (BPI) | X | X | X | X |
| Perceived mental and physical fatigue | Multidimensional Fatigue Inventory (MFI-20) | X | X | X | X |
| Sleep Quality | Pittsburg Sleep Quality Index (PSQI) | X | X | X | X |
| *Physical activity and function* | | | | | |
| Physical activity | ActivePAL triaxial accelerometer (PAL) | X | X | X | X |
| Physical capacity | 1-minute sit to stand (1-min-STS) | X | X | X | X |
| Leg muscle endurance | 30-second sit to stand (30-sec-STS) | X | X | X | X |
| Short physical performance battery | Guralnic test, 3m gait speed, 5-times sit-to-stand (SPPB) | X | X | X | X |
| Muscle strength upper extremity | Hand-grip strength (Jamar dynamometer) | X | X | X | X |
| Data registrations and expense indicators | Adherence to intervention/self-maintenance | | X | X | X |
| | Adverse events recorded | | X | X | X |
| | Number of hospital admission | past 12-mo | X | X | X |
| | Length of stay–hospital admission | past 12-mo | X | X | X |
| | Consultant visits | past 12-mo | X | X | X |
| | Mortality | | X | X | X |
| | Comorbidities | X | X | X | X |
| **Descriptive variable** | | | | | |
| Lung function test | FEV1/FVC ratio in % | X | | | X |
| | FEV1 (% predicted) | X | | | X |
| | DLCO (% predicted) | if available | | | if available |
| Anthropometries | Body Mass Index (BMI) | X | X | X | X |
| | Body weight (kg) | X | X | X | X |
| | Body height (cm) | X | X | X | X |
| | Fat Free Mass Index (FFMI) | if available | if available | if available | if available |
| Other information reported | Smoking status | X | X | X | X |
| | Medication prescribed | X | X | X | X |
| | Bone fractures | past 12-mo | X | X | X |
| | BODS | X | | | X |
| | Charlson's Comorbidity Index (CCI) | X | | | X |

Table 9 presents outcomes and flowchart of measurements in the study.

score ranging from 1–5, the highest score corresponds to highest level of fatigue. A total of 100 is possible. A systematic review investigated fatigue in patients with COPD and found a fatigue scale cut-off score of $\geq 13$ (severe fatigue) [54]. Within early stages of Parkinson's disease, a screening cut-off score for general fatigue of 11 points and for total MFI of 60 points for clinically significant fatigue has been suggested [55]. In this study we will use a cut-off $\geq 13$ points.

*Pittsburg Sleep Quality Index (PSQI)* is used to assess sleep quality [56]. The PSQI measures sleep disturbances and sleep habits within the past month. The PSQI consists of 11 items that

are scored on a four-item likert scale ranging from 0–3. In the present study questions 1–9 are answered by the patient. A global PSQI score of $\geq$ 5 reliably distinguishes "poor" quality sleepers from "good" quality sleepers (diagnostic sensitivity 89.6% and specificity 86.5%) [56, 57].

**Physical activity.**   To measure 24-h physical activity patients will be asked to wear an activePAL[TM] triaxial accelerometer (PAL Technologies Ltd., Glasgow, UK) [58]. The activePAL will be worn for 5 consecutive days prior to randomization, 5 days after completion of 10-week intervention, 5 days after completion of 35-weeks follow-up assessment and 5 days after completion of 75-weeks follow-up assessment. In this study the level of physical activity is an exploratory variable. Approximately 50% of the included patients will be asked to wear an activePAL[TM] and these patients will preferably be living within a 25 km distance of Hvidovre University Hospital for practical reasons. The activePAL[TM] accelerometer measures time spent sitting/lying, standing, walking (number of steps, cadence) and number of transitions (i.e. sit-to-stand), steps per day, metabolic equivalent of task (MET) and time in bed. The activePAL[TM] is a valid and reliable tool used in several COPD studies to track and measure physical activity [59, 60]. Demeyer et al have suggested a distribution-based MCID of 600–1100 steps per day for patients with COPD, this range has been found consistent by Driver and colleagues [61, 62]. We consider an increase of 600 step per day as clinically important change for this study considering the target group included.

**Physical capacity and function.**   *1-minute sit to stand test (1-min-STS)* is used to measure physical capacity. The 1-min-STS test is performed using an armless chair with a height of 44–46 cm, the patient arms are crossed and placed at the chest. If the patient is not able to perform sit-to-stand without the use of arms, the scoring is 0. The patient is asked to perform 1–2 practice attempts before performing the test for 1 min. The patient is instructed to perform as many as possible sit-to-stand at a self-selected pace. The patient will be given a notification when 15 sec. are left of the test, otherwise no other communication will be provided. The patient is allowed to take a pause during the test, but the time will still be ongoing.

1-min-STS have been chosen because of its relationship to submaximal efforts of physical capacity. Furthermore, the test is possible to implement in a home environment. McDonald and colleagues found that 1-min-STS and 6 min. walking test were strongly correlated ($r$ = 0.61) and that the test demonstrated satisfactory validity and responsiveness [63]. A MCID of 2.5 repetitions have been suggested [64, 65].

*30-second sit to-stand (30-sec-STS)* is used to measure leg muscle strength. 30-sec-STS is a common test for assessing physical capacity for COPD patients [66]. The test is practical and simple in a clinical setting. The 30-sec-STS test is performed using an armless chair with a height of 44–46 cm. Patients are asked for perform as many as possible within 30 sec. No communication is provided during the test. If the patient is not able for perform sit-to-stand without support from the chair, the score is 0. The number of repetitions from the 30-sec-STS will be retrieved during performance of the 1-min-STS. The patients are not aware of this mid-time (30-sec-STS) retrieval when performing the 1-min-STS. A MCID of $\geq$ 2 has been suggested by Zanini and colleagues [67].

*Short Physical Performance Battery and (SPPB)* is used to measure physical function and frailty [68]. The test consists of a) standing balance test (Guralnik), b) a 3-meters-walk test and, c) a 5-sit-to-stand. The balance test is performed with three separate tests: stance side-by-side for 10 sec., stance semi-tandem for 10 sec., and tandem stand for 10 sec. A maximum score of balance is 30. The 3-meters-walk test is performed after the balance test. In the gait test the patient walks in her/his normal pace. The time used is recorded and the test is performed twice. If the patient uses a walking aid this is noted. The 5-sit-to-stand test (5-STS) is performed twice with 2 min. apart. It is recorded how fast a patient can perform 5-sit-to-stand without using their arms. If the patient is not able to perform the test without support, then the

score is 0 points. The total score of all tests is comprised to a final SPPB score with a possible score of 0 to 12 points [69]. A scoring of 0–7 point is defined as physical frail, 8–9 points physical pre-frail and 10–12 physical non-frail. A change of 0.5 points on the SPPB is considered a small meaningful change, while a change of 1 point on the SPPB is considered a substantial meaningful change within an older community-dwelling population [70, 71].

*Hand-grip strength* (Jamar dynamometer) is used to measure muscle strength in the upper extremity [72]. The Hand-grip strength (HGS) is measured on both sides in a seated position. The patient keeps her/his arm slightly abducted, the elbow flexed at 90 degrees and the forearm in a neutral position between supination and pronation. All measurements are repeated three times in each side, and the individual average are calculated and used for analysis. To our knowledge no MCID exists for HGS in patient with COPD. Pragmatically, we will consider a percentual strength increase of 15% relative to the peak baseline HGS as clinically relevant when interpretating changes over time.

**Descriptive variables.** Lung function test, medication for COPD, age, sex, arthrometries (weight, height, body mass index), marital status, length of education, smoking status, years with COPD, and Charlson comorbidity index (CCI). Furthermore, Body-mass index, airflow Obstruction, Dyspnea and Exericse (BODE index) are registered, and it is a multidimensional scoring system and capacity index used for patients with COPD. The BODS index is developed from the original BODE index using 5-STS and the index demonstrate prognostic value for predicting mortality risk for patients with COPD [73]. The lung function test is conducted at the respiratory department of the referral hospitals by a pulmonologist or respiratory nurse before baseline assessment. All hospitals in the study have clinically approved spirometry equipment.

**Other variables.** Adverse events, hospitalizations and deaths will be extracted from Statistic Denmark throughout the trial. Data extraction include number of outpatient visits, number of hospitalizations, hospitalization days and number of deaths. Data will be analyzed as all-cause, respiratory related admissions and outpatient visits. Self-reported falls and bone-fractures during the intervention will be -recorded.

**Adverse events.** Any adverse events (AE) during testing and during training will be recorded in the CRF and training protocol, respectively. A standardized method will be used; event description (i.e., a clear identification of the AE), onset dates (dates for when the even began and resolved), intensity (classification of the AE's severity–e.g., mild, moderate, severe) and causality assessment (evaluation of the relationship of the AE to the study, e.g., probably related, possibly related, not related). To minimize bias the investigators are trained to ask neutral questions about the AE's during patient interview to ensure data collection is as objective as possible.

Serious AEs are reported to the principal investigator within 24 hours. A pulmonologist, a respiratory nurse and a clinical physiotherapist will survey and evaluate serious AEs.

**Dropouts.** All dropouts and reasons for these will be recorded by the research team. Dropout will occur when a patient informs the investigators that he/she is unwilling to participate in the study any longer. If a patient does not participate in an exercise session, without cancellation, the patient is contacted to establish reason for not attending. In cases where it is not possible to contact the patient, he/she will be recorded as drop-out after 3 months (from the maintenance program, unless there is a valid reason for not attending the sessions, for instance hospitalization, disease or family circumstances). If a patient is withdrawn from one of the intervention groups, all equipment will be collected. The patient will still be eligible for follow-up assessments, if the patient wishes to, and data will be analyzed according to the intention-to-treat analysis.

**Change to initial plan.**  As per February 2024 we have started a collaboration with the municipality of Copenhagen, Center for pulmonary diseases. The center will begin recruitment in the early spring of 2024. We plan to establish collaboration to other surrounding municipalities close to Bispebjerg and Hvidovre University Hospitals.

Furthermore, a Facebook site has been launched with the purpose of recruiting patients with COPD that are not referred to the hospital, but where the general practitioner has the overall treatment responsibility.

**Publication.**  The intervention study will be disseminated in two separate publications. A 10-weeks (primary endpoint) RCT effect study and subsequently a publication on the effects from a 65-weeks maintenance program (that is 75 weeks from baseline, secondary endpoint).

Furthermore, we plan to publish the following: A) An original qualitative article with focus on data from a mixed-method study using qualitative interviews and a generic survey to gain insight into patients' thoughts, experience and attitudes towards new rehabilitation models and the maintenance program. B) Reliability calculations. Intra- and inter tester reliability will be calculated on physical outcome measures (1min-STS and 30-sec-STS, handgrip strength, SPPB) in fifty consecutively recruited patients. The retest is completed seven to 10 days after baseline assessment and prior to intervention start. The reliability in the patient reported questionnaires (CAT, HADS, EQ-5D, PSQI, MFI and BPI) will be calculated in the same sample of 50 patients. The reliability for the physical outcome measures and patient reported outcomes will be published in separate articles. C) An original article with focus on CUA of the two new delivery models for PR targeting patients with COPD who opt out of outpatient PR service. D) Other hypothesis generating explorative articles with focus for example on sleep, fatigue, dose-response between physical activity and PR, responder- and non-responder analysis.

## Discussion

This study protocol describes a multicenter RCT, which aims to investigate the effects of two new rehabilitation models and compare the clinical benefits with usual care after 10-weeks and subsequently investigate the effect and adherence of an 65-week exercise maintenance program.

To our knowledge this is the first RCT study that investigates new rehabilitation models used specifically to target the group of patients with COPD, unable to particpate in center-based programs. Secondly, this is the first RCT study to head-to-head compare the equivalence from two well-established homed-based models, that is group-based PTR and indiviual HPR.

The research interventions are designed to meet different motivations and overcome the barriers for rehabilitation activity among patients with COPD. As the two exercise programs are designed to involve the same expenses and staff time per person, the healthcare-sectorrelated costs of the programs are equivalent and anticipated to represent a subsequent scale-up in the setting of hospitals and municipalities. Thus, costs will not make one of the regimes (PTR vs HPR) more attractive than the other, and the choice with regard to implementation can be informed by other aspects, such as coherence with other initiatives in the given municipality and the ability to individualize approaches.

If the investigators can show that the two new delivery models, including motivational maintenance programs supported by a tablet with a video conference system, reduce symptoms, and increase QoL and adherence equally but superior to usual care, then the REPORT study will provide new evidence. This will allow patients with COPD the choice of a validated, effective, simple and safe approach to PR and maintenance exercise that meet individual wishes, preferences, needs and barriers towards participation in beneficial rehabilitation and maintenance programs. The results from the REPORT study will have a scaling up potential

and could encourage patients with COPD throughout Denmark and internationally to participate and exercise more due to increased accessibility.

## Supporting information

**S1 Protocol.**
(PDF)

**S1 Checklist. SPIRIT 2013 checklist: Recommended items to address in a clinical trial protocol and related documents\*.**
(DOC)

**S1 File.**
(PDF)

## Acknowledgments

We thank all the participating departments of respiratory medicine, municipalities, departments of physiotherapy and occupational therapy for their involvement and making the necessary resources available and the conduction of this RCT trial possible. The steering committee for valuable discussions during the protocol development. Furthermore, we thank the Danish Lung Foundation, Telemedical Center Regional Capital Copenhagen, TrygFonden, Danish Association for Physiotherapists, Jascha Fonden, Skibsreder Per Henriksen, R og Hustrus fond, Amager-Hvidovre Hospital Forskningspulje and Lundbeck Fonden for their founding support.

## Author Contributions

**Conceptualization:** Christina Nielsen, Nina Godtfredsen, Stig Molsted, Charlotte Ulrik, Thomas Kallemose, Henrik Hansen.

**Supervision:** Henrik Hansen.

**Validation:** Nina Godtfredsen, Stig Molsted, Charlotte Ulrik, Thomas Kallemose, Henrik Hansen.

**Visualization:** Christina Nielsen, Stig Molsted, Charlotte Ulrik, Thomas Kallemose, Henrik Hansen.

**Writing – original draft:** Christina Nielsen, Nina Godtfredsen, Stig Molsted, Charlotte Ulrik, Thomas Kallemose, Henrik Hansen.

**Writing – review & editing:** Christina Nielsen, Nina Godtfredsen, Stig Molsted, Charlotte Ulrik, Thomas Kallemose, Henrik Hansen.

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
