## [Decision Letter · Decision Letter 0]

12 Jul 2024

PONE-D-24-17848

Supervised pulmonary tele-rehabilitation and individualized home-based pulmonary rehabilitation for patients with COPD, unable to participate in center-based programs. The protocol for a multicenter randomized controlled trial - the REPORT study.

PLOS ONE

Dear Dr. Nielsen,

Thank you for submitting your manuscript to PLOS ONE. After careful consideration, we feel that it has merit but does not fully meet PLOS ONE’s publication criteria as it currently stands. Therefore, we invite you to submit a revised version of the manuscript that addresses the points raised during the review process.

We look forward to receiving your revised manuscript.

Kind regards,

Tim Alex Lindskou

Academic Editor

PLOS ONE

 [This research project has received specific grants from the Danish Lung Foundation (Charitable funding), Telemedical Center Regional Capital Copenhagen (Governmental funding), TrygFonden (Charitable funding), Danish Association for Physiotherapists (Charitable funding), Jascha Fonden (Charitable funding), Skibsreder Per Henriksen, R og hustrus fond (Charitable funding), Amager-Hvidovre Hospital Forskningspulje (Governmental funding) and Lundbeck Fonden (Charitable funding). The Grants covers expenses conducting the trial, salary for the project employed, and the University fees for the PhD education for Christina Nielsen. 

I have uploaded funding letters from TrygFonden (for Henrik Hansen - last author) and for myself from AHH Forskningspulje 2023 and 2023 along with Skibsreder Per Henriksen og Hustrus Fond. ].  

3. In the online submission form, you indicated that [No datasets were generated or analysed during the current study. All relevant data from this study will be made available upon study completion.]. 

Additional Editor Comments (if provided):

Reviewers' comments:

Reviewer's Responses to Questions

**Comments to the Author**

1. Does the manuscript provide a valid rationale for the proposed study, with clearly identified and justified research questions?

Reviewer #1: Yes

Reviewer #2: Yes

2. Is the protocol technically sound and planned in a manner that will lead to a meaningful outcome and allow testing the stated hypotheses?

Reviewer #1: Yes

Reviewer #2: Yes

3. Is the methodology feasible and described in sufficient detail to allow the work to be replicable?

Reviewer #1: Yes

Reviewer #2: Yes

4. Have the authors described where all data underlying the findings will be made available when the study is complete?

Reviewer #1: No

Reviewer #2: Yes

5. Is the manuscript presented in an intelligible fashion and written in standard English?

Reviewer #1: Yes

Reviewer #2: Yes

6. Review Comments to the Author

You may also provide optional suggestions and comments to authors that they might find helpful in planning their study.

Reviewer #1: In this study protocol, a multicenter three-arm randomized controlled trial is being proposed to investigate the change in respiratory symptoms associated with the COPD Assessment Test after 10-weeks. Additionally, the study aims to test the equivalence of the two intervention arms and their superiority to the control arm.

Minor revisions:

1- Line 580: The standard statistical term for average is mean.

2- Line 587: Indicate the underlying covariance structure that will be used in the mixed effects models or the criteria for selecting it.

3- Line 596: Clarify that the differences in proportions are being tested, rather than the number/frequency. Group differences in the proportion of patients ...

4- Line 779: Consider replacing “registered” with “recorded”.

5- Line 781: Indicate if adverse events will be collected according to a standardized method.

7- Line 801: Typographical error: Change “have” to “has”.

Reviewer #2: Minor comments and questions to the Authors:

1. How are you going to assess the cognitive impairment? What criteria for exclusion are you going to apply?

2. Is PTR also a home-based procedure? What is the location of PTR? The differences between PTR and HPR are described in detail, but main differences are lost in the load of information, what makes it difficult for a non-expert to understand. Would a consise table with comparison feasable?

3. Resasons of resignation from conventional PR should be assessed in all patients. They may be different and could cause a bias in the final assessment. Similarly, the randomisation should be performed taking into account the severity of functional impairment (initial FEV1, DLCO, hyperinflation), and muscle strengh. Alternatively, these data should be taken into consideration in statistical evaluation.

7. PLOS authors have the option to publish the peer review history of their article (what does this mean?). If published, this will include your full peer review and any attached files.

Reviewer #1: No

Reviewer #2: **Yes: **Wojciech Piotrowski

---

## [Author Response · Author response to Decision Letter 0]

12 Aug 2024

Response to Reviewers 

Hvidovre Hospital, 8th of August 2024

Dear Academic Editor and Reviewers 

Thank you for reviewing our manuscript and for your comments. We are very glad for your comments as it will improve our manuscript. 

Concerning journal requirements from Academic Editor: 

Ad. 1. The manuscript has been checked to meet the PLOS ONE’s style requirements (main body and title, authors affiliations). 

Ad. 2. The funders had no role in design etc., this in now stated in the revised cover letter and in the article (please see line 40-41). 

Ad. 3. Dataset. 

The authors are not able to provide our datasets for public use due to ethical laws and very strict juristic data protection law in Denmark. Any possible access or sharing demands a part application to; (1) Danish Data Protection Agency, (2) Ethics Committee of the Capital Region, (3) National Health Data Authorities. Only if the applications are approved data will be considered available for sharing. The authors will not be able to support this process and a prolonged process must be expected.

Ad. 4. Ethics statement have been moved to the Method section and deleted in other section (please see line 285-293). 

Ad. 5. Separate captions for each figure are now included in our manuscript. 

Ad. 6. The reference list has been reviewed. I have found some discrepancies, a duplicate of same article reference. This have been corrected. The reference list does not contain any references to retracted articles. 

Concerning Reviewers comments: 

Ad. 1. No revisions made. 

Ad. 2. No revisions made. 

Ad. 3. No revisions made. 

Ad. 4. Concerning data policy. 

Data access in Denmark are under very strict juristic data protection law. Any possible access or sharing demands a part application to; (1) Danish Data Protection Agency, (2) Ethics Committee of the Capital Region, (3) National Health Data Authorities. Only if the applications are approved data will be considered available for sharing. The authors will not be able to support this process and a prolonged process must be expected.

Ad. 5. No revisions made. 

Ad. 6. 

# Reviewer 1: 

- 6.1: Change have been made (please see line 643)

- 6.2: Change have been made by our statistician (please see line 652) 

- 6.3.: Change have been made (please see line 659)

- 6.4.: Change have been made (please see line 847)

- 6.5.: Extra information have been provided in the manuscript (please see line 851-856)

- 6.7.: Change have been made (please see line 880)

#Reviewer 2: 

- 6.1.: We do not assess cognitive impairment with any standardized tests in our clinical practice and in other studies conducted at our research department. As described on page 10, line 241 in the manuscript, the clinical impression/assessment is based on the judgment of an outpatient respiratory nurse and a respiratory physician who refer the patient to the research study. Additionally, in cases of doubt concerning patient eligibility, such as cognitive impairment or comorbid conditions, the project's responsible respiratory physician and principal investigator are consulted for advice, and a pre-meeting is planned, after which the final decision is made. Finally, if the patient is unable to understand and follow the instructions and surveys during the baseline/inclusion assessment, the blinded assessor will contact the study's responsible PI for a prompt decision on whether to proceed or exclude the patient, with reasons provided.

We have specified this process in more detail in the section “Eligibility Criteria.” We hope this clarifies your question. Please see page 10, lines 243-248.

- 6.2.: Concerning PRT home-based procedure and location of PTR – this is now more clearly stated in the manuscript.

Concerning the differences of the two groups – PTR and HPR – an extra short and concise table have been uploaded in the manuscript. Thank you for your recommendation. 

- 6.3.: Concerning reasons for resignation and/or declining PR, this information is registered in a study database for further use in the final assessment. This information is now added in the manuscript (please see line 243-246). 

With regards for randomization and severity of functional impairment, the statistical analyses will 

include adjustment for FEV1, among others. Unfortunately, we are not able to obtain data on DLCO 

and/or hyperinflation as this is not a standard examination in Danish outpatient clinics unless a pulmonologist deems it necessary. It is not possible for the investigators to change the randomization method as recruitment has begun and are ongoing. Furthermore, FEV1 is only used as a descriptive value and not an effect value. 

In our author group we have discussed that our section concerning blinding needed more details. We have therefore added information, please see line 319-326. Furthermore, I have found a type error (in the section concerning 1-min STS test) and I have corrected it. 

I hope that the abovementioned is satisfactory otherwise please do not hesitate to contact us again and thank you for reviewing our manuscript. 

With best regards

Christina Nielsen

---

## [Decision Letter · Decision Letter 1]

14 Oct 2024

Supervised pulmonary tele-rehabilitation and individualized home-based pulmonary rehabilitation for patients with COPD, unable to participate in center-based programs. The protocol for a multicenter randomized controlled trial - the REPORT study.

PONE-D-24-17848R1

Dear Dr. Nielsen,

We’re pleased to inform you that your manuscript has been judged scientifically suitable for publication and will be formally accepted for publication once it meets all outstanding technical requirements.

Kind regards,

Esedullah Akaras

Academic Editor

PLOS ONE

Additional Editor Comments (optional):

Reviewers' comments:

Reviewer's Responses to Questions

**Comments to the Author**

1. Does the manuscript provide a valid rationale for the proposed study, with clearly identified and justified research questions?

Reviewer #1: Yes

Reviewer #2: Yes

2. Is the protocol technically sound and planned in a manner that will lead to a meaningful outcome and allow testing the stated hypotheses?

Reviewer #1: Yes

Reviewer #2: Yes

3. Is the methodology feasible and described in sufficient detail to allow the work to be replicable?

Reviewer #1: Yes

Reviewer #2: Yes

4. Have the authors described where all data underlying the findings will be made available when the study is complete?

Reviewer #1: No

Reviewer #2: Yes

5. Is the manuscript presented in an intelligible fashion and written in standard English?

Reviewer #1: Yes

Reviewer #2: Yes

6. Review Comments to the Author

You may also provide optional suggestions and comments to authors that they might find helpful in planning their study.

Reviewer #1: All comments have been adequately addressed.

Reviewer #2: -6.1.: We do not assess cognitive impairment with any standardized tests in our clinical

practice and in other studies conducted at our research department. As described on

page 10, line 241 in the manuscript, the clinical impression/assessment is based on

Powered by Editorial Manager® and ProduXion Manager® from Aries Systems Corporation

the judgment of an outpatient respiratory nurse and a respiratory physician who refer

the patient to the research study. Additionally, in cases of doubt concerning patient

eligibility, such as cognitive impairment or comorbid conditions, the project's

responsible respiratory physician and principal investigator are consulted for advice,

and a pre-meeting is planned, after which the final decision is made. Finally, if the

patient is unable to understand and follow the instructions and surveys during the

baseline/inclusion assessment, the blinded assessor will contact the study's

responsible PI for a prompt decision on whether to proceed or exclude the patient, with

reasons provided.

We have specified this process in more detail in the section “Eligibility Criteria.” We

hope this clarifies your question. Please see page 10, lines 243-248.

COMMENT: I accept that

-6.2.: Concerning PRT home-based procedure and location of PTR – this is now more

clearly stated in the manuscript.

Concerning the differences of the two groups – PTR and HPR – an extra short and

concise table have been uploaded in the manuscript. Thank you for your

recommendation.

COMMENT: Thank you for the table, it helps to understand your research

-6.3.: Concerning reasons for resignation and/or declining PR, this information is

registered in a study database for further use in the final assessment. This information

is now added in the manuscript (please see line 243-246).

With regards for randomization and severity of functional impairment, the statistical

analyses will

include adjustment for FEV1, among others. Unfortunately, we are not able to obtain

data on DLCO

and/or hyperinflation as this is not a standard examination in Danish outpatient clinics

unless a pulmonologist deems it necessary. It is not possible for the investigators to

change the randomization method as recruitment has begun and are ongoing.

Furthermore, FEV1 is only used as a descriptive value and not an effect value.

In our author group we have discussed that our section concerning blinding needed

more details. We have therefore added information, please see line 319-326.

COMMENT: thank you, however the incomplete functional data are, in my opinion, a disadvantage

Thank you for your responses and adjustments. I accept the article in the present form.

7. PLOS authors have the option to publish the peer review history of their article (what does this mean?). If published, this will include your full peer review and any attached files.

Reviewer #1: No

Reviewer #2: **Yes: **Wojciech Piotrowski

---

## [Editor Report · Acceptance letter]

29 Oct 2024

PONE-D-24-17848R1 

PLOS ONE

Dear Dr. Nielsen, 

I'm pleased to inform you that your manuscript has been deemed suitable for publication in PLOS ONE. Congratulations! Your manuscript is now being handed over to our production team.

Kind regards, 

on behalf of

Dr. Esedullah Akaras 

Academic Editor

PLOS ONE